

# Effect of boundary layer low-level jet on fog fast spatial propagation

Shuqi Yan[1], Hongbin Wang[1,*], Xiaohui Liu[2], Fan Zu[1], Duanyang Liu[1,*]

[1]Key Laboratory of Transportation Meteorology of China Meteorological Administration, Nanjing Joint Institute for Atmospheric Sciences, Nanjing, 210041, China
[2]Merchant Marine College, Shanghai Maritime University, Shanghai, 201306, China

*Correspondence to*: Hongbin Wang (kaihren@163.com); Duanyang Liu (liuduanyang2001@126.com)

**Abstract.** The spatiotemporal variation of fog reflects the complex interactions among fog, boundary layer thermodynamics and synoptic systems. Previous studies revealed that fog can present **fast spatial propagation** feature and attribute it to boundary layer low-level jet (BLLJ), but the effect of BLLJ on fog propagation is not quantitatively understood. Here we analyze a large-scale fog event in Jiangsu, China from 20 to 21 January 2020. Satellite retrievals show that fog propagates from southeast coastal area to northwest inland with the speed of 9.6 m/s, which is three times larger than the ground wind speeds. The ground meteorologies are insufficient to explain the fog fast propagation, which is further investigated by WRF simulations. The fog fast propagation could be attributed to the BLLJ occurring between 50 and 500 m, because the wind speeds (10 m/s) and directions (southeast) of BLLJ core are consistent with fog propagation. Through sensitive experiments and process analysis, three possible mechanisms of BLLJ are revealed: 1) The abundant oceanic moisture is transported inland, increasing the humidity of boundary layer and promoting condensation; 2) The oceanic warm air is transported inland, enhancing the inversion layer and favouring moisture accumulation; 3) The moisture advection probably promotes upper-level fog formation, and later it subsides to ground by turbulent mixing of fog droplets. The fog propagation speed would decrease notably by 6.4m/s (66%) if the BLLJ-related moisture and warm advections are turned off.

## 1. Introduction

Fog is a kind of low-visibility weather phenomenon that occurs at near surface, causing adverse impacts on traffic transportation. The formation, development and dissipation of fog are the comprehensive results of the interactions among radiation, moisture, microphysics, turbulence, aerosols and other factors (Gultepe et al., 2007; Koračin et al., 2014; Nakanishi, 2000). The relations of fog with meteorological factors are highly variable under different conditions. Therefore, the mechanism of fog evolution needs to be intensively studied.

Under favourable conditions, the fog intensity or its spatial extent can develop extraordinarily fast with time. Field observations conducted at single site reveal that visibility in fog can deteriorate drastically, from about 1km to less than 200m within 30min (Li et al., 2019). It is referred to as fog burst reinforcement, which is firstly raised by Korb et al. (1970) and systematically reviewed by Liu et al. (2012) and Li et al. (2019). Fog burst reinforcement is accompanied by the drastic formation of fog droplets, sudden increase of fog liquid water and broadening of droplet spectrum (Liu et al., 2017; Liu et al., 2021). Additionally, fog can develop rather fast in spatial extent, i.e., the **fast spatial propagation of fog** (Zhu et al., 2022). It is reflected by the successive visibility dropping in space along a certain direction. The influencing factors of fast



spatial propagation could be more complex than that of the burst reinforcement at single site, which have received fewer
quantitative studies recently.
Synoptic systems and planetary boundary layer (PBL) thermodynamic structures are key to understanding the cause of fog
burst reinforcement and fast propagation. Weak cold air invasion and radiative cooling is an important factor for fog burst
reinforcement and fast propagation (Liu et al., 2011; Wang et al., 2020). Dhangar et al. (2021) demonstrated that the radia-
tive cooling at surface and fog top can increase supersaturation and promote fog vertical development. Shen et al. (2022)
found that the different cooling rates at two nearby stations lead to a remarkable difference in fog formation time, fog du-
ration and vertical extent. Sufficient water supply is also an important factor. Wobrock et al. (1992) revealed that the role
of moisture advection outweighs radiative cooling in large-scale fog events. Pu et al. (2008) found that two layers of mois-
ture advection enhance fog development and maintenance. Under stable synoptic systems, the PBL thermodynamic can
also favour fog burst reinforcement and fast fog propagation. The formation of dense fog is usually accompanied by strong
inversion layer, of which the intensity could reach 16K/100m (Pu et al., 2008; Liu et al., 2012). Liu et al. (2016) found that
upper-level warm advection and low-level cold advection significantly enhance inversion intensity and promote fog de-
velopment. The vapor advection resulting from southerly winds further increases fog intensity. Appropriate turbulence also
facilitates fog formation and enhancement (Ye et al., 2015). Turbulent results in the exchange of heat and moisture within
PBL, e.g., the downward entrainment of vapor and cold air can promote condensation and droplet formation (Liu et al.,
2016; Zhang et al., 2005). Other studies highlight the role of hygroscopic aerosols and aerosol indirect effects in strong fog
events (e.g., Boutle et al., 2017; Quan et al., 2021; Wang et al., 2023; Yan et al., 2021).
Previous studies find that the large-scale fog events are accompanied by boundary layer low-level jet (BLLJ), and try to
attribute the spatial propagation of fog to BLLJ. The causes of BLLJ include such as synoptic systems, terrian effect and
inertial oscillation (Kraus et al., 1985). Tian et al. (2019) demonstrated that the warm-and-wet southerly BLLJ favours wa-
ter vapor transportation and inversion layer construction, and later the fog is triggered by a weak cold front invasion. Wu et
al. (2020) found that strong northerly BLLJ associated with cold air can destroy inversion layer and lead to early dissipa-
tion of fog, while weak BLLJ can promote fog maintenance. Li et al. (2012) revealed that the strengthened turbulence gen-
erated by BLLJ wind shear promotes vertical mixing and facilitates fog development. However, the relations between
BLLJ and fog propagation and the key synoptic factors have not been quantitatively addressed. Also, the current horizontal
and vertical observations are not sufficient to reveal the mechanism of fog propagation. It requires further investigation by
numerical models.
In this work, we study a large-scale fog event with fast propagation feature occurring in Jiangsu Province, China from 20
to 21 January 2020. By combination of observations and numerical simulations, we aim to quantitatively reveal the BLLJ
effect on fast fog propagation to and identify the key impact factors and mechanisms. This work is expected to better un-
derstand the complex interactions among synoptic systems, PBL thermodynamics and fog spatial propagation, as well as
provide prediction indicators for operational fog forecast. The study is organized as follows: Section 2 describes the data,
methods and numerical models of this study. Sections 3.1 to 3.4 analyze the fog propagation feature and PBL characteris-
tics. Section 3.5 quantitatively study the BLLJ effect on fast fog propagation and identifies key influencing factors. Section
4 concludes the findings of this study.





## 2.    Data, methods and model configuration

### 2.1 Data and study area

This study focuses on the Jiangsu area, China (Figure 1), where a large-scale fog event occurred from 20 to 21 January 2020. We collected the data from 70 ground automatic weather stations (AWS) in Jiangsu Province, China. The data is recorded by every 10 minutes, including visibility, temperature, relative humidity (RH), wind direction and wind speed. This data is used to analyze the temporal variation of meteorology, as well as evaluate the model performance on temperature, RH and wind.

The geostationary satellite Himawari 8 (https://www.eorc.jaxa.jp/ptree/index.html) is used to retrieve nighttime fog area and evaluate the model performance of fog simulation. The high spatiotemporal resolution (2km in space and 1h in time) is suitable for detecting the fast evolution of fog area. This satellite observation includes 16 bands, and the bands at 3.9 and 11.2 μm are used.

The ERA5 reanalysis data (https://cds.climate.copernicus.eu/cdsapp#!/dataset/reanalysis-era5-pressure-levels) is used to analyze synoptic conditions and provide initial & boundary fields for model simulation. The grid resolution is 0.125° (about 12.5km) and the time interval is 6h. All the time in this study is local time (UTC+8).

### 2.2 Methods

#### 2.2.1 Satellite fog retrieval

Since the ground AWS stations are not sufficiently fine in spatial resolution, the high spatiotemporal resolution product of Himawari 8 is suitable to study the propagation of fog. Nighttime fog has notable different optical properties at the bands of 3.9μm and 11.2μm, so it can be indicated by the dual-band brightness temperature difference ($Tbb_{3.9}$ minus $Tbb_{11.2}$) lower than a threshold (Cermak et al., 2008). In this study, the threshold is determined to be -2 K following the dynamic threshold algorithm proposed by Di Vittorio et al. (2002). Daytime fog after 08:00 is not retrieved because we mainly focus on the formation and development stage of fog before 08:00.

#### 2.2.2 Fog propagation speed calculation

We calculate the propagation speed according to satellite retrieved fog area. At 22:00 on 20 January 2020, a tiny fog area appeared at Nantong and Yanchen coastal region with an area smaller than 50km2 (figure not shown). The center of this fog area is set as point A (120.6°E, 32.9°N). We draw a line starting from A with an arbitrary direction, and find its intersection with the fog boundary area at 07:00 next day (point B). Then the propagation speed in this direction can be calculated by the distance from A to B divided by 9 hours (22:00~07:00). By looping from 0 to 360 with the interval of 1°, propagation speeds in all directions are calculated, and the maximum speed is defined as the fog propagation speed.

The fog propagation speed is verified by AWS data. We select three representative stations along the fog propagation direction, Dafeng (DF; 120.48°E, 33.20°N, 14m), Baoying (BY; 119.30°E, 33.23°N, 15m), Sihong (SH; 118.22°E, 33.48°N,





13m) (Figure 1). According to their distances and the time differences when visibility drops to 200m, the propagation
speed between two adjacent stations is calculated.
2.2.3 Process analysis on fog
The simulated fog is indicated by fog liquid water content (LWC). Process analysis is used to quantify the contribution of
each physical process to LWC variation (Schwenkel et al., 2019; Yan et al., 2020). The variation of LWC is related to the
following terms:
$$\frac{\partial \text{LWC}}{\partial t} = \underbrace{-\left(u\frac{\partial}{\partial x}+v\frac{\partial}{\partial y}+w\frac{\partial}{\partial z}\right)\text{LWC}}_{\text{Advc}} + \left(\frac{\partial \text{LWC}}{\partial t}\right)_{\text{Vmix}} + \left(\frac{\partial \text{LWC}}{\partial t}\right)_{\text{Cond}} + \left(\frac{\partial \text{LWC}}{\partial t}\right)_{\text{Sedi}} + \left(\frac{\partial \text{LWC}}{\partial t}\right)_{\text{other}}$$

where Advc includes horizontal and vertical advection, Vmix is associated with the fog droplet vertical exchange by tur-
bulent mixing, Cond is the vapor condensation (negative means droplets evaporation), Sedi is fog droplets sedimentation.
Other microphysical processes include autoconversion, accretion and cold phase processes. They are much smaller than the
previous four processes, so they can be safely ignored.
**2.3   Model configuration and experiments**
The Weather Research and Forecasting model (WRF) is implemented to study the fast spatial propagation of fog events.
Two domains are set up (Figure 1). The parent domain covers East China, with the grid size of 181×181 and grid interval
of 9 km. The nested domain covers Jiangsu Province and its coastal area, with the grid size of 199×199 and grid interval of
3 km. To simulate the turbulent process more reasonably, the vertical levels are refined to 42 levels, with 25 levels under
1500m and 9 levels under 100m (Yang et al., 2019; Yan et al., 2020). The first model level is about 4m. The model is driv-
en by the initial and boundary field from ERA5 Reanalysis. The simulation starts at 08:00 on 19 January and ends at 08:00
on 21 January 2020, with the first 24h as spin-up period. All the time in this study is local time (UTC+8).
Fog simulation is sensitive to the choice of parameterization schemes (Steeneveld et al., 2014; van der Velde et al., 2010).
Through massive tests, the QNSE boundary layer scheme (Sukoriansky et al., 2005) and Pleim-Xiu land surface scheme
(Pleim et al., 2009) yield the best simulation performance. Other parameterization schemes are listed in Table 1. The simu-
lated fog is indicated by the liquid water content (LWC) greater than 0.015g/kg under the height of 500m, which corre-
sponds to horizontal visibility less than 1km (Kunkel, 1983).
Apart from the base experiment, three sensitive experiments are performed to elucidate the mechanism of fast fog propa-
gation (Table 1). The experiment "Tadv0" turns off the temperature advection within PBL during the fog period. The ex-
periment "QvAdv0" and "QcAdv0" are the same as "Tadv0" except that turning off water vapor advection and fog water
advection, respectively. The experiment "NoAdv" turns off all the advections above. Therefore, the differences of the base
experiment with Tadv0, QvAdv0, and Qcadv0 represent the effect of temperature advection, moister advection, and fog
water advection, respectively. The reasons and results of the sensitive experiments will be discussed in Section 3.5.



## 3.    Results and discussions

### 3.1    Fog overview and synoptic background

The studied fog event occurs at the night of 20 January and dissipates in the daytime of 21 January 2020 (Figure 2). Figure 3 shows the synoptic situations at 08:00 and 20:00 on 20 January. At 500hpa, a frontal zone is located north of 38°N. The Jiangsu area is dominated by prevailing westerly flows with no obvious troughs. At 850hpa, a ridge moves eastward and controls Jiangsu area. The descending motions associated with the ridge and the nocturnal radiative cooling at ground favour the establishment of inversions. At ground level, a weak cold high pressure moves eastward with the central pressure of 1030hpa. The Jiangsu area is dominated by uniform pressure field with small wind speeds, which strengthens atmospheric stratification stability and promotes the accumulation of aerosols and moisture. The moisture condition in Jiangsu is additionally favoured by the water vapor transportation from ocean by easterly winds at 20:00. Under this conductive situation, the fog event occurred from nighttime of 20 to daytime of 21 January over Jiangsu Province (Figure 2).

### 3.2    Fog and ground meteorology variation

Hourly Himawari 8 satellite image clearly shows the spatial propagation of fog (Figure 2). The fog initials at 22:00 on 20 January in Nantong and Yanchen coastal region with an area smaller than 50km$^2$. Later, this small fog area expands to a large-scale fog. Specifically, the southeast side of fog area varies relatively slowly, but the northwest side expands remarkably, indicating a large propagation speed. At 07:00 on 21 January, the front of fog expands to Anhui Province. After 07:00, the fog begins to dissipate (figure not shown). Figure 4 quantitatively describes the propagation direction and speed of fog. From the east to south directions (the fourth quadrant), fog propagation speed is less than 3m/s. In the west-northwest and west directions, fog propagation speed is larger than 6m/s, and the maximum propagation speed is 9.6m/s occurring at 160° direction. The fast propagation of fog is also reported previously in Jiangsu area (Gao et al., 2023; Zhu et al., 2022), where the fog propagates from coastal area to west boundary of Jiangsu within about 10h.

Visibilities at three representative stations, Dafeng (DF), Baoying (BY) and Sihong (SH) are used to verify the fog propagation speed calculated by satellite (Table 2; Figure 5). At DF, fog forms (visibility less than 1km) early at 19:45 on 20 January. The visibility drops sharply at 23:15 and reaches the minimum at about 00:15. At BY and SH, fog forms in turn, and their visibilities also have burst decreasing feature at 03:40 and 07:00, respectively. We calculate the fog propagation speed by the distances among stations and the time differences when visibility drops to 200m. The propagation speed is 7.6 m/s between DF and BY and 8.3 m/s between BY and SH. These values correspond to the speed calculated by satellite observation.

Figure 5 shows the variation of other meteorological fields. We focus on the characteristics from fog formation to the burst visibility dropping (indicated by yellow dashed lines). At DF, the northerly wind decreases to lower than 1.5m/s at fog formation, which causes the weak cold advection and temperature decreasing. The temperature keeps decreasing and favours the burst reduction of visibility at 23:15. The vapor content (indicated by dew point) increases sharply before 17:00 and decreases slightly since then, so the RH increasing after fog formation is caused by temperature drop. At BY and SH, the wind directions are dominantly southeast and the speeds are generally less than 2m/s before fog formation. The tem-



perature keeps decreasing and vapor content keeps increasing, leading to the further reduction of visibility. Later, the
southeasterly winds obviously enhance by about 1m/s, which may contribute to the burst visibility dropping due to the in-
tensified vapor advection from ocean.
The preliminary cause of fog formation and intensification are summarized. As located near the ocean, the moisture at DF
reaches the maximum prior to fog formation, so the fog formation and intensification are largely caused by radiative cool-
ing and weak cold advection. At BY and SH, the temperature cooling rate is weaker than DF, which is partly due to the
weak warm advection by southeasterly winds. The vapor advection by southeasterly winds favours fog development, and
the burst decrease in visibility coincides with the increase in wind speed. Therefore, deduced from BY and SH, the vapor
transportation associated with southeasterly winds could be an important reason for northwesterly propagation of fog.
However, it is obvious that the ground wind speed is rather small compared with fog propagation speed. Statistics on AWS
stations show that although wind direction (east, southeast and south winds at 70% stations) is generally in accordance
with fog propagation direction, wind speed is lower than 3m/s at 97% stations from 22:00 to 07:00, which is about
one-third of the fog propagation speed. Therefore, the ground meteorological field is insufficient to explain the fast propa-
gation of fog. The fog PBL characteristics and the key influencing factors need to be investigated by numerical simula-
tions.

### 3.3  Model evaluation

Figure 6 evaluates the model performance on temperature, relative humidity (RH) and wind field at surface. The simulated
temperature and RH agree well with observations, with the root mean square error (RMSE) of 1.0K and 11%, respectively.
The simulation reasonably captures the wind direction transition from north to east, and the RMSE is less than 1m/s.
Figure 2 compares the satellite observed and simulated fog area. The simulation is only evaluated before 07:00, because
the dissipation of fog after 08:00 is not the focus in this study. The model reasonably captures the spatiotemporal evolution
of fog, with a slight overestimation of 5~10% in fog area.
Overall, the simulation reasonably captures the temporal variation of meteorology and reproduces the spatial propagation
of fog. It establishes the basis for discerning the mechanism of fog propagation.

### 3.4  Characteristics of fog and PBL structure

The thermodynamic variation of PBL is crucial for understanding the propagation of fog. Figure 7a shows the temporal
variation of horizontal winds in vertical directions. The simulated wind speed is consistently smaller than 4m/s under about
30m, while it remarkably increases with height. At 18:00 on 20 January, a large wind speed zone (>6m/s) forms at the
height between 50 and 500m in the east of 120°E. Since then, the large wind zone moves westward quickly accompanied
by wind speed increasing. During the fog period, the average wind speed exceeds 6m/s at the height between 50 to 500m
(Figure 7b), which is commonly larger than the wind speed in most fog events. Here, we refer to this large wind speed
zone as **boundary layer low-level jet (BLLJ)**. The existence of BLLJ is supported by ERA5 reanalysis on 1000hpa and
975hpa levels (Figure 7b).




The formation of BLLJ is likely caused by the easterly movement of a high pressure at 1000hpa over East China. The cen-
tral pressure gets enhanced, which strengthens the pressure gradient over Jiangsu area and favours wind speed increasing
(figure not shown). The jet core (maximum wind speed) occurs at about 1000hpa (200m), with the time-averaged speed of
10m/s (Figure 7b). At that level, the dominant wind direction is southeast and the wind speed over fog area is 8~16m/s
(Figure 7c), which can fit the propagation direction and speed of fog. Also, the expansion speed of vertical fog zone is
comparable to the movement speed of jet core (Figure 7a). Therefore, we hypothesize that the southeasterly BLLJ could
account for the fast propagation of fog.
Previous studies reveal that southerly BLLJ can transport abundant water vapor to China inland and thus promote fog for-
mation (Liu et al., 2016; Tian et al., 2019). Figure 8 shows the temporal variation of water vapor mixing ratio (Qv) profiles.
Since the vapor content over the ocean is higher, it is transported to inland areas by southeasterly BLLJ. The BLLJ can
further increases the Qv in PBL by wind speed horizontal convergency and vertical shear. The larger wind speed in BLLJ
zone and lower wind speed outside BLLJ zone cause wind speed convergence, which favours the increase in PBL moisture.
Additionally, the turbulence generated by vertical shear of wind speed can promote vapor turbulent mixing, leading to the
higher Qv above surface being entrained downward and increasing the ground Qv (Gao et al., 2007). The Qv under 300m
is generally higher than 3g/kg under the effect of BLLJ. Wu et al. (2020) also found that BLLJ continuously transports
water vapor to fog layer, resulting in surface Qv higher than 3g/kg. It is notable that the expansion of vertical fog area co-
incides with the movement of the zone of Qv>4g/kg. Therefore, moister advection by BLLJ could be an important reason
for fast fog propagation.
BLLJ is reported to result in warm advection and deepen inversion layer previously (Tian et al., 2019), and inversion layer
is an important reason for fog burst reinforcement in most fog cases (e.g., Li et al., 2019; Liu et al., 2012; Jiao et al., 2016).
Figure 9 shows the temporal variation of temperature profile and inversion layer. The inversion layer here refers to the
height above ground where temperature monotonically decreases with height. Since 20:00 on 20 January, the ground tem-
perature keeps decreasing due to radiative cooling. Within the fog area, the temperature drop is more significant, which is
due to the longwave radiative cooling by fog droplets (Bott, 1991; Jia et al., 2018). Approximately above the fog top, there
is an obvious warm air mass transported from ocean to inland areas. The BLLJ-induced warm advection increases vertical
temperature gradient and strengthens atmospheric stability. Accordingly, the inversion height over non-fog areas basically
keeps increasing. The approximate inversion layer height is about 100~300m, with the maximum inversion intensity of
15K/100m. Such a strong inversion is also reported in a dense fog event (16K/100m) by Pu et al. (2008). It favours the
accumulation of vapor and condensation nuclei, which is also a possible reason for fog formation in the downstream area.
Additionally seen from Figure 9, the west boundary of vertical fog region below about 100m has a negative slope, i.e., fog
forms at upper level ahead of forming at ground. The height at which fog firstly forms is shown in Figure 10. An initial fog
area forms at ground level before 00:00 on 21 January. Since then, the majority of fog area firstly forms at upper level
(about 10~66m) over the downstream area, while the ground fog in downstream area forms about 0~20min later than up-
per-level fog. The formation of upper-level fog may also be caused by the BLLJ-induced moisture advection. In addition,
the fog water advection (Section 2.2.3) to downstream area by BLLJ could also be a potential reason. We hypothesize that
the formation of ground fog is partly favoured by the subsidence of upper-level fog. Stratus-lowering or upper-fog subsid-



ence to ground has been reported by previous studies (e.g., Haeffelin et al., 2010; Liu et al., 2012); the base height of stra-
tus can be smaller than 100m before fog formation (Dupont et al., 2012; Fathalli et al., 2022), which is basically close to
our results (10~66m in Figure 10). While in this event, the upper-fog subsidence remains to be verified by additional
high-spatiotemporal resolution vertical observations.
According to above results, three potential factors for fog propagation are raised: BLLJ-related temperature advection,
moisture advection and fog water advection. These advections possibly promote fog formation in the upper level, and sub-
sequently the upper-level fog could subside to ground by the turbulent mixing or sedimentation of fog droplets. Currently,
their contributions to fog propagation have not been quantitatively revealed. Therefore, it will be addressed in the next sec-
tion.

### 3.5  Quantitative reasons for fast fog propagation

Four sensitive experiments, Tadv0, QvAdv0, QcAdv0 and NoAdv0 (Section 2.3) are conducted to quantify the respective
contributions of temperature advection, moister advection, fog water advection and all these advections to fog propagation
(Figure 11). Under the condition with no advections (Figure 11a-d), there is a 80% decrease in fog area and a 6.4m/s (66%)
decrease in propagation speed, which highlights the role of BLLJ-related advections. When turning off temperature advec-
tion (Tadv0) (Figure 11e-h), the original fog area in the base experiment shrinks 50% in size and breaks into separate fog
patches, and the propagation speed decreases by about 5.2m/s (54%). When turning off moisture advection (QvAdv0)
(Figure 11i-l)., the fog area shrinks by 62% in size and the propagation speed decreases by about 4.6m/s (48%). When
turning off fog water advection (QcAdv0) (Figure 11m-p), the fog area nearly keeps unchanged during 00:00~04:00 and
decreases moderately in size (about 25%) at 06:00 The propagation speed decreases moderately by 2.4m/s (25%). Deduced
from the changes in fog area and propagation speed under various experiments, we can infer that the BLLJ-related warm
and moisture advection, especially moisture advection, could be the major cause of fast spatial propagation, while fog wa-
ter advection has a minor contribution.
We further perform process analysis on LWC (Section 2.2.3) to illustrate the mechanism of fog propagation (Figure 12).
The horizontal and vertical values of Advc and Sedi are at least one order of magnitude smaller than that of Cond and Sedi,
indicating that fog water transportation to downstream areas and droplet sedimentation to ground are not the causes of fog
propagation. At 00:00 on ground level, Cond is positive over the newly formed fog area (blue and cyan colors surrounding
the fog area), indicating that fog firstly forms at ground by radiative cooling before 00:00. After 02:00, Cond is almost
negative over the entire fog area, indicating that fog does not firstly form at the ground level (otherwise Cond would have
positive values). The formation of ground fog may be contributed by the LWC turbulent entrainment from upper level,
because Vmix shows significant positive values after 02:00. In the vertical direction, Vmix and Cond are still two domi-
nant physical processes (Figure 12b), and their signs show opposite patterns. At lower level (0~30m), Cond is negative and
Vmix is positive, which is the same as their ground characteristics. At upper level (30~200m), Cond is positive and Vmix
is negative instead, indicating that fog water is produced by vapor condensation at upper level and then being entrained to
ground. The significant positive Cond supports that BLLJ-related moisture advection promotes vapor condensation and fog





formation at upper level, and the significant positive Vmix may indicate that the upper-level fog favours ground fog for-
mation by turbulent exchange of LWC.
Figure 13 summarizes the mechanism of fog propagation. During the nighttime, a southerly BLLJ controls the study region,
and the jet core intensity is about 10m/s which occurs at about 200m. The ground fog propagates northwestward with the
speed of 9.6m/s. The BLLJ favours the fast fog propagation by three possible mechanisms: 1) BLLJ transport sufficient
vapor from ocean to inland area. The turbulence strengthened by wind speed shear further moistens the PBL and promotes
vapor condensation. This could be the dominant mechanism. 2) BLLJ transports warmer air from ocean to inland area and
deepens the inversion layer. The strengthened inversion favours the accumulation of vapor and condensation nuclei. 3) The
strong moisture advection could promote the upper-level fog formation in the downstream area, and later it subsides to
ground by turbulent exchange of fog droplets. The subsidence of upper-level fog to ground needs to be verified by addi-
tional observations.
The results could facilitate the understanding of cloud formation and development. Clouds, such as convective clouds, can
develop and expand extraordinarily fast under strong synoptic forcing or unstable conditions. Fog can be viewed as a kind
of near-surface stratus cloud, which usually forms under stable conditions with weak synoptic forcings. However, as re-
vealed in this study, it can also develop and propagates fast under the effect of BLLJ. The quantitative relations between
BLLJ and fog fast propagation may have implications on the cloud formation and development mechanism under stable
synoptic conditions.
## 4. Conclusions
Previous studies have found that the spatial propagation of fog could be rather fast under favourable conditions, and the
boundary layer low-level jet (BLLJ) could be a potential reason. In this study, we analyze the fast spatial propagation fea-
ture of a large-scale fog event in Jiangsu Province, China by high spatiotemporal resolution ground and satellite observa-
tions. The key impact factors and mechanisms of the BLLJ effect on fast spatial propagation are quantitatively revealed by
WRF model simulations. Results show that:
The fog initials at 22:00 on 20 January 2020 over Jiangsu coastal area, and it reaches the west boundary of Jiangsu at 07:00
next day. Satellite retrievals show that the southeast side of fog area varies slightly but the northwest side expands fast,
with the maximum propagation speed of 9.6m/s. During the fog period, the ground wind direction is consistent with fog
propagation, which favours the vapor transportation from ocean and promotes fog formation. However, the wind speed
(<3m/s) is at least one-third less than the fog propagation speed. Therefore, the ground meteorologies are insufficient to
explain the fast propagation of fog. The influencing factors and mechanisms need to be investigated by exploring the PBL
characteristics through numerical simulations.
The WRF model well simulates the temporal variation of meteorologies and reproduces the spatiotemporal evolution of
fog area. A BLLJ (>6m/s) exists at the height between 50 and 500m. The jet core occurs at 1000hpa (200m) with the
southeasterly winds of 10m/s, which can fit the propagation direction and speed of fog. Therefore, the southeasterly BLLJ



is hypothesized to be the cause of fast propagation. BLLJ creates favourable PBL conditions by transporting moisture and
warm air from ocean. The moisture advection and the vapor turbulent mixing generated by wind speed shear increase the
humidity within PBL, and the propagation of fog area coincides with the movement of high humidity zone (vapor mixing
ratio>4g/kg). The warm advection from ocean deepens inversion layer and additionally favours the accumulation of mois-
ture and condensation nuclei. Additionally, it is found that fog could firstly form at upper layer and subsides to ground
within 0~20min. The moisture advection is also responsible for the formation of upper-level fog.
Sensitive experiments quantitatively reveal the contributions of moisture advection and temperature advection to fog
propagation. When moisture (temperature) advection is turned off, the fog area decreases by 62% (50%) and the propaga-
tion speed decrease by about 4.6m/s (5.2m/s). Process analysis on fog liquid water content (LWC) further illustrates the
mechanism of fog propagation. Condensation (Cond) and LWC turbulent exchange (Vmix) are two important physical
processes. At upper level (30~200m), Cond is positive and Vmix is negative. It indicates that BLLJ-related moisture ad-
vection significantly promotes condensation and probably favours fog formation at upper level. At ground and lower level
(0~30m), Cond is basically negative and Vmix is positive. It indicates that fog droplets at upper level are entrained down-
ward by turbulent mixing, leading to the subsequent formation of ground fog. The subsidence of upper-level fog to ground
needs to be verified by additional observations.
In this study, by combination of observations and simulations, we have revealed the effect of southeasterly BLLJ on fog
propagation, and quantified the contributions of BLLJ-related moisture advection and temperature advection to fog propa-
gation. Three possible mechanisms are concluded: 1) Moisture advection from ocean promotes vapor condensation in
downstream area, which could be the dominant cause; 2) Warm advection from ocean deepens inversion layer and addi-
tionally promote vapor accumulation within PBL. 3) The moisture advection probably promotes upper-level fog formation
first, and later it subsides to ground by turbulent mixing of fog droplets. The coexistence of fast fog propagation and BLLJ
is not a common phenomenon, so finding more cases requires additional work. It should be addressed in future studies in
order to deeply understand the relationships between fog propagation and BLLJ under different regions and synoptic con-
ditions. Their quantitative relationships could facilitate the understanding of cloud formation and development under stable
synoptic conditions, since fog can be viewed as near-surface stratus cloud that can potentially propagate fast under stable
conditions.

*Code and data availability*. Some of the data repositories have been listed in Section 2. The other data, model outputs and
codes can be accessed by contacting Duanyang Liu via liuduanyang2001@126.com.
*Author contributions*. SY performed the model simulation, data analysis and manuscript writing. HW and DL proposed the
idea, supervised this work and revised the manuscript. XL helped the revision of the manuscript. FZ provided and analyzed
the observation data.
*Competing interests*. The authors declare that they have no conflict of interest.
*Acknowledgements*. This work is supported by the Special Project of Innovative Development of CMA (CXFZ2023J022),



Open Research Foundation of Jiangsu Marine Meteorology (HYQX2022), Beijige Foundation (BJG202307), Research
Foundation of Jiangsu Meteorology Bureau (KM202307), Basic Research Fund of CAMS (2022Y025).

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



Table 1.   Model parameterization schemes and sensitive experiments

| Physical scheme | Option |
|---|---|
| Boundary layer | QNSE |
| Microphysics | Lin double moment |
| Longwave radiation | RRTM |
| Shortwave radiation | Goddard |
| Land surface | Pleim-Xiu |
| Cumulus | Grell-3D |
| Grid nudging | Off |
| Observation nudging | Off |
| Experiment | Description |
| Base | The base condition |
| Tadv0 | Turning off temperature advection |
| QvAdv0 | Turning off water vapor advection |
| QcAdv0 | Turning off fog water advection |
| NoAdv | Turning off all advections above |


Table 2.   The times when visibility reaches 1000m, 500m and 200m at three representative stations. (DF:Dafeng,
BY:Baoying, SH:Sihong).

| Station | Location | Formation (Vis=1000m) | | Vis=500m | | Vis=200m | |
|---|---|---|---|---|---|---|---|
| | | Time | Wind | Time | Wind | Time | Wind |
| DF | 120.48°E,33.20°N | 19:45 | 1.3m/s, E | 22:55 | 1.2m/s, E | 23:45 | 1.3m/s, E |
| BY | 119.30°E,33.23°N | 01:25 | 1.2m/s, ESE | 03:15 | 1.4m/s, ESE | 03:45 | 1.3m/s, SE |
| SH | 118.22°E,33.48°N | 04:50 | 1.6m/s, ESE | 06:10 | 1.3m/s, ESE | 07:15 | 2.4m/s, ESE |
| | Distance (km) | Time difference (h) | | Time difference (h) | | Time difference (h) | |
| DF-BY | 110 | 4.7 | | 4.3 | | 4.0 | |
| BY-SH | 105 | 3.4 | | 2.9 | | 3.5 | |







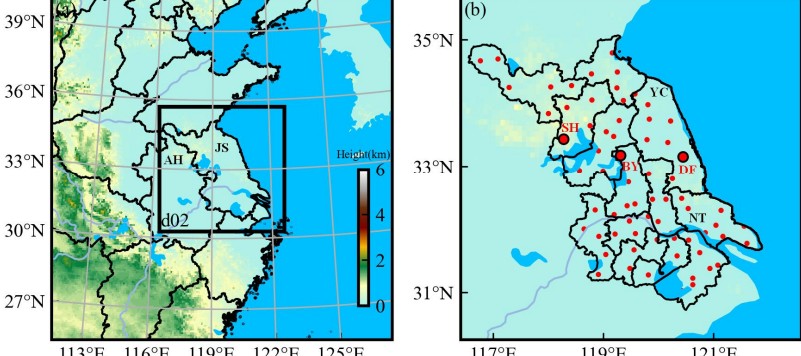


Figure 1.   The parent and nest model domain. The shaded color is terrain height. The red points are automatic weather stations
in Jiangsu, China; and the three larger points are Sihong (SH), Baoying (BY), and Dafeng (DF) stations. The black labels are
some province or city names. (JS:Jiangsu Province; AH:Anhui Province; YC:Yanchen; NT:Nantong).

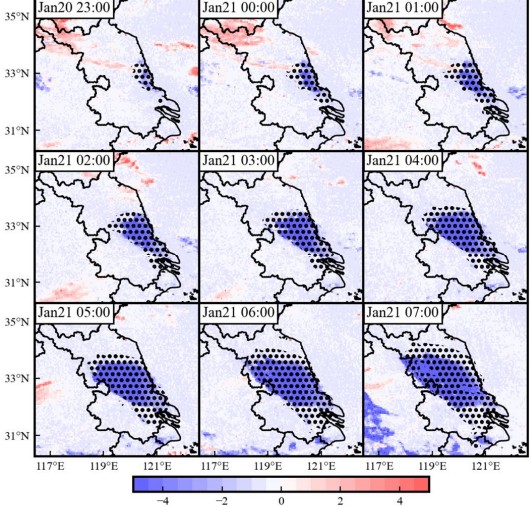


Figure 2.   The spatial evolution of fog. The black dots are simulated fog areas. The shaded colors are satellite observed bright-
ness temperature difference (3.9μm minus 11.2μm), where the blue colors (smaller than -2 K) indicate the fog areas.



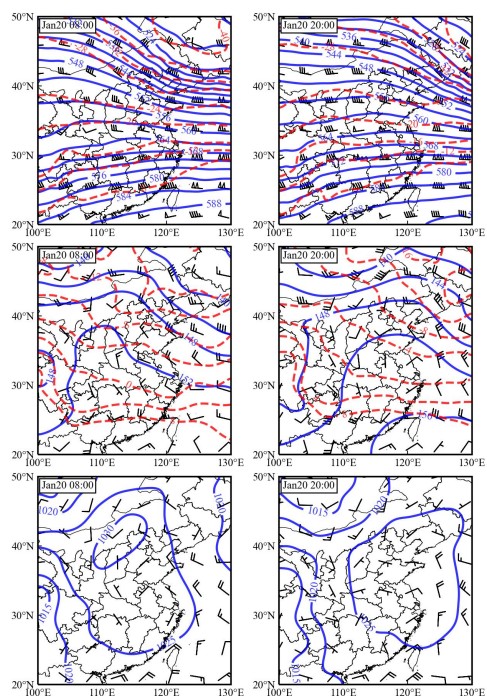


Figure 3.   The synoptic background of 500hpa (first row), 850hpa (second row) and surface (third row) at 08:00 and 20:00 on
20 January 2020.

462



463

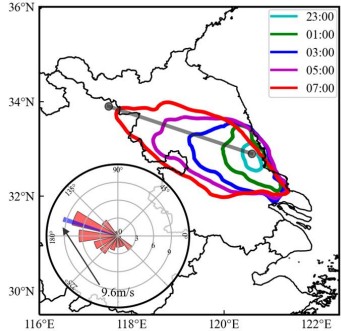

464

Figure 4. The colored curves are the fog boundaries (satellite retrievals) from 23:00 on 20 January to 07:00 next day every 2
hours. The gray straight line indicates the fog propagation direction, and the vertical features of meteorologies at this line will be
analyzed in Figures 7, 8, and 9. The lower-left polar plot is the fog propagation speed at 16 directions (22.5° interval), and the
narrow blue bar highlights the maximum propagation speed (9.6m/s) occurring at 160° direction (north-northwest).

469

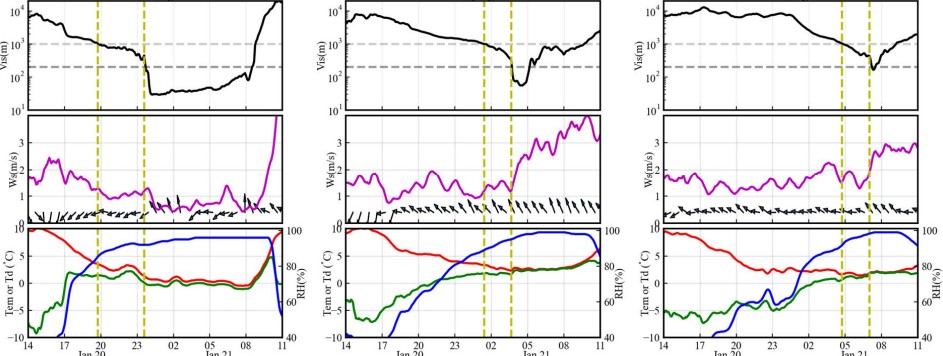

470

Figure 5. The temporal variation of ground visibility (Vis; black line), wind speed (Ws; pink line), wind direction (vectors),
temperature (Tem; red line), dew point (Td; green line) and relative humidity (RH; blue line) at Dafeng, Baoying, and Sihong
stations. The horizontal dashed lines are visibilities of 1000m and 200m. The vertical dashed lines mark the times of fog for-
mation and visibility burst dropping.





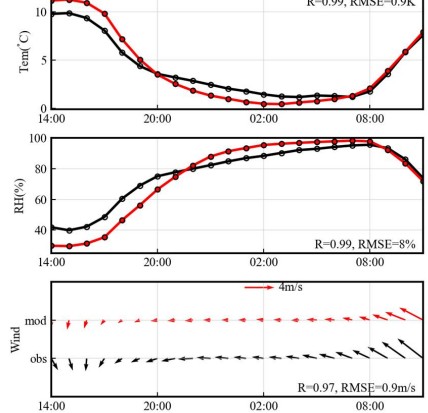


Figure 6.   The model performance on 2m Temperature (Tem), 2m Relative humidity (RH) and 10m wind speed and direction.
The red color is simulation and black color is observation. The time is from 14:00 on 20 January 2020 to 11:00 next day.


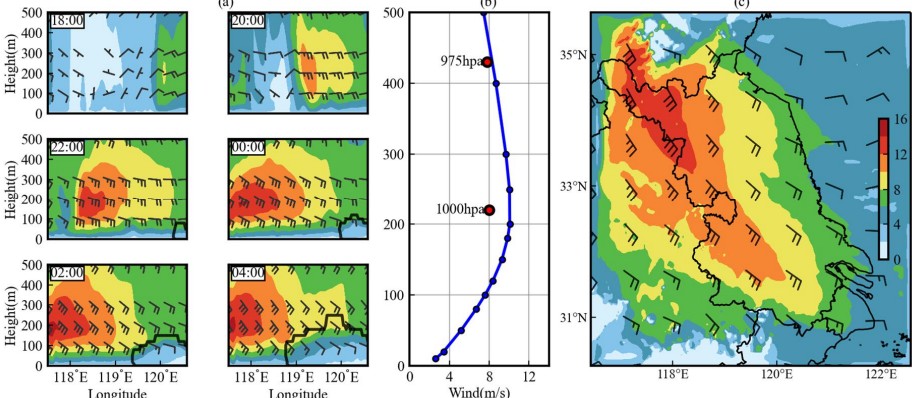


Figure 7.   (a) The height-longitude variation of horizontal wind direction (vectors) and wind speed (shaded colors) at the cross-
ing line in Figure 4. The lower-right black polygons are the fog area. The times are from 18:00 on 20 January to 04:00 next day.
(b) The averaged wind speed profile at the crossing line during 23:00~07:00. The two red points are the wind speed calculated
from ERA5 reanalysis. (c) The averaged wind direction (vectors) and wind speed (shaded colors) at 1000hpa during
23:00~07:00.

487



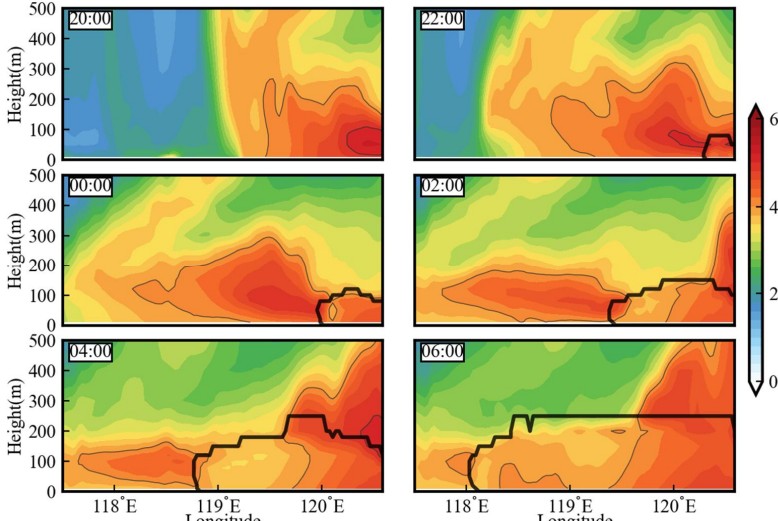

Figure 8.   The height-longitude distribution of water vapor mixing ratio (g/kg) at the crossing line in Figure 4. The deep black polygons are the fog area. The light black lines are the region of water vapor mixing ratio larger than 4g/kg. The times are from 20:00 on 20 January to 06:00 next day.



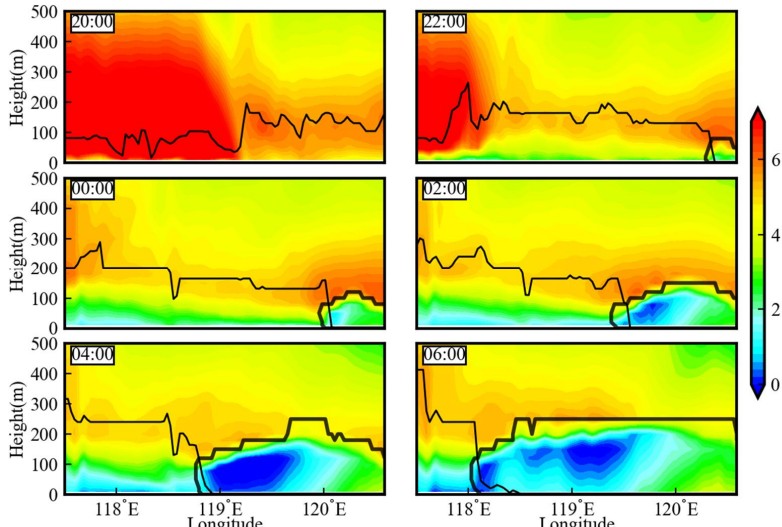

Figure 9. Same as the previous figure, but for the temperature. The bold black polygons are the fog area. The thin black lines are the top of inversion layer.

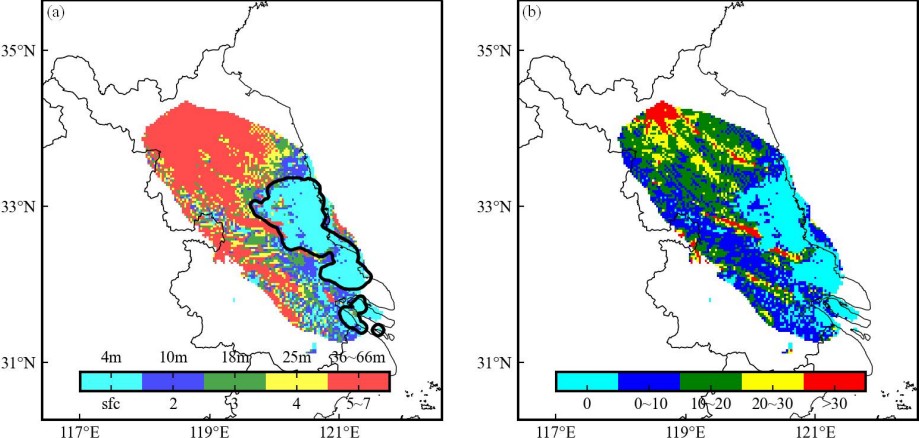

Figure 10.   (a) The height (shaded color) at which fog firstly forms. The black contours are the ground fog areas at 00:00 on 21
January 2020. The colorbar represents the model level and the corresponding height above surface. For example, the cyan colors
indicate that fog firstly forms at the surface level with the corresponding height of about 4m. The red colors indicate that fog
firstly forms at the 5th to 7th model level with the corresponding height of about 36~66m. (b) The time differences between
ground fog formation and upper-level fog formation. For example, the cyan colors indicate that fog firstly forms at ground. The
blue colors indicate that the ground fog forms 0~10min later than the upper-level fog formation.



506

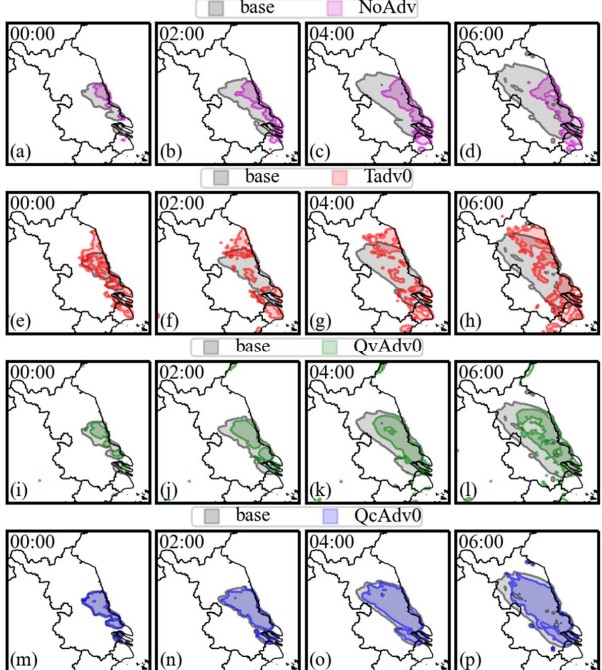

507

Figure 11.    The temporal variation of ground fog area under different experiments from 00:00 to 06:00 on 21 January. The black color is the base experiment. The Tadv0 (red), QvAdv0 (green) and QcAdv0 (blue) are the experiments turning off temperature advection, moisture advection and fog water advection, respectively. The NoAdv (pink) is the experiment turning off all of the above advections.



513

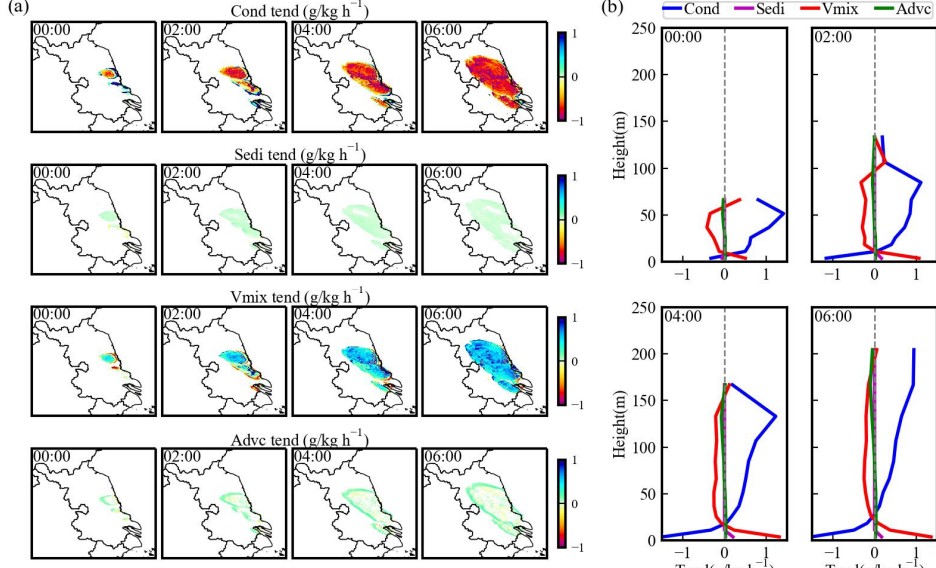

514

Figure 12.    (a) The spatial distribution of the four process tendencies contributing to LWC variation at ground level. (b) The

vertical profiles of the process tendencies averaged in fog area. The times are from 00:00 to 06:00 on 21 January.

(Cond:condensation or evaporation; Sedi:sedimentation; Vmix:turbulent exchange; Advc:horizontal and vertical advection).






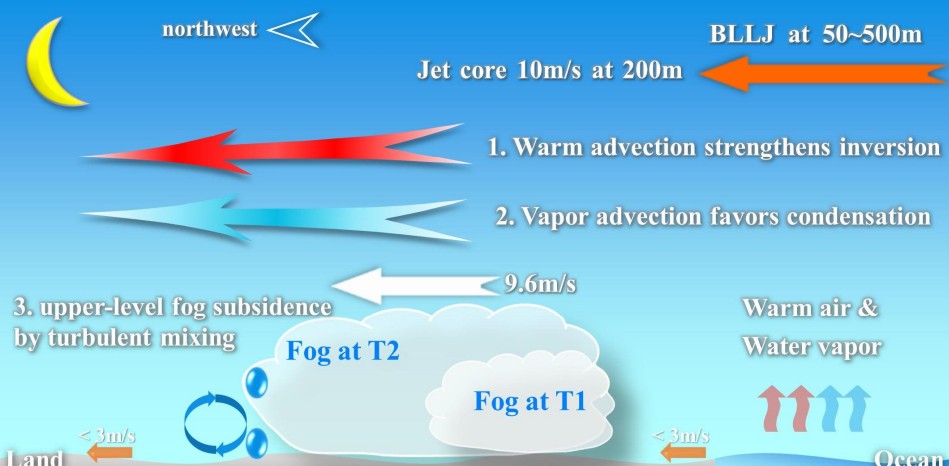


Figure 13. The concept diagram of fog propagation. The ground wind speed (short orange arrows) is generally less than 3m/s.
A southeasterly BLLJ exists at the height from 50 to 500m, and the jet core intensity is 10m/s at 200m (the long orange arrow).
The updraft arrows represent the warm and wet air from ocean. The two cloud shapes are fog areas at two adjacent times, and the
white arrow indicates the fog propagation speed (9.6m/s). The fog propagation is probably caused by three approaches: 1) Mois-
ture advection from ocean promotes vapor condensation in the downstream area, which could be the dominant cause (the blue
fancy arrow); 2) Warm advection from ocean deepens inversion layer and additionally promotes vapor accumulation within PBL
(the red fancy arrow); 3) The moisture advection probably result in the upper-level fog formation, and later it subsides to ground
by turbulent mixing of fog droplets (the blue water drops and circular arrows).