# Peer review of "Effect of boundary layer low-level jet on fog fast spatial propagation"

_EGUsphere, 2023_

## Author Comment (AC1)

**Response to community reviewer**

Dear Dr. Zhou,

Thanks for your constructive comments and suggestions on our manuscript (egusphere-2023-1299). We have studied them carefully and made revisions on the manuscript. These comments and the corresponding replies are listed below.

The reviewer's comments are highlighted by gray. Followed by the comments are our responses. The texts led by **"In Section xx"** are the current texts in manuscript. The added texts or  are colored by red.

With regards,

Shuqi Yan

**Comments:**

1. The authors claimed that the WRF model well simulated the atmospheric status of the fog event. But this conclusion was based only on comparison of simulated variables to the surface observations. Just surface environment well simulated is not enough. The high level atmospheric fields should also be well simulated since this study focuses on the influence of high level wind jets on the fog event. Please add such a comparison of the model with high level fields. This can gives readers more confidence to the results and conclusions.

Thanks for this suggestion. We have added the model evaluation on vertical temperature, humidity and winds within PBL by collecting data from sounding stations. The nearest sounding station to the fog area is Sheyang (SY, 120.25, 33.76, 3m).

Overall, the model reasonably simulates the temperature, humidity and wind profiles (Figure 6b). The detailed discussions are added into the manuscript.

> **In Section 2.1**
> This study focuses on the Jiangsu area, China (Figure 1), where a large-scale fog event occurred from 20 to 21 January 2020. We collected the data from 70 ground automatic weather stations (AWS) in Jiangsu Province, China. [data description……]. Additionally, the Sheyang (SY; 120.25°E, 33.76°N; 3m) station is a sounding station that used for model evaluation in the vertical direction. The sounding observations include temperature, RH, wind direction and wind speed which are sampled each second. It is conducted twice a day (00UTC and 12UTC).

[Figure]

Figure 1. The parent and nest model domain. The shaded color is terrain height. The red points are automatic weather stations in Jiangsu, China. The three larger circle points are Sihong (SH), Baoying (BY), and Dafeng (DF) stations, and the square point is Sheyang (SY) sounding station. The black labels are some province or city names. (JS:Jiangsu Province; AH:Anhui Province; YC:Yanchen; NT:Nantong).

**In Section 3.3**

Figure 6b evaluates the model performance on temperature, RH and wind field in the vertical direction at SY sounding station. The temperature profile is simulated well by the model, with the mean bias of less than 1K. The RH bias is relatively small below about 200m, while it is a bit larger above 200m at 08:00 on 21 January. The simulated wind speed and direction are basically consistent with observation. The large winds (greater than 6m/s) at about 200m are well reproduced by the model, indicating that the model reasonably simulates boundary layer low-level jet. Studies on boundary layer low-level jet are presented in next sections.

[Figure]

Figure 6b. The model performance on temperature (red), RH (blue) and wind (barbs) profiles at Sheyang sounding station. For temperature and RH, the observations are scatters and simulations are solid lines. For wind barbs, the left column is observations and the right column is simulations. The scatters and barbs are interpolated onto 0~600m every 100m.

2. In this study, simulated LWC was used to investigate the fog development during the fog event. The LWC equation included several important items such as condensation part, sedimentation part, vertical mixing part, etc. Then how to identify these items in the study? I don't remember there are items in WRF's output files. These items were calculated or diagnosed from the WRF outputs?

The LWC equation includes advections, condensation, sedimentation, and vertical mixing as you mentioned. These terms are already calculated by WRF physical processes, which can reasonably reflect the variation of LWC in fog events. They can be outputted by the model after modifying codes by users.

The terms controlling LWC variation have been already presented in Section 2.2.3.

**In Section 2.2.3**

The simulated fog is indicated by fog liquid water content (LWC). Process analysis is used to quantify the contribution of each physical process to LWC variation (Schwenkel et al., 2019; Yan et al., 2020). The variation of LWC is related to the following terms:

$$\frac{\partial \text{LWC}}{\partial t} = \underbrace{-\left(u\frac{\partial}{\partial x}+v\frac{\partial}{\partial y}+w\frac{\partial}{\partial z}\right)\text{LWC}}_{\text{Advc}} + \left(\frac{\partial \text{LWC}}{\partial t}\right)_{\text{Vmix}} + \left(\frac{\partial \text{LWC}}{\partial t}\right)_{\text{Cond}} + \left(\frac{\partial \text{LWC}}{\partial t}\right)_{\text{Sedi}} + \left(\frac{\partial \text{LWC}}{\partial t}\right)_{\text{other}}$$

where Advc includes horizontal and vertical advection, Vmix is associated with the fog droplet vertical exchange by turbulent mixing, Cond is the vapor condensation (negative means droplets evaporation), Sedi is fog droplets sedimentation. Other microphysical processes include autoconversion, accretion and cold phase processes. They are much smaller than the previous four processes, so they can be safely ignored.

3. This study only focused on the fog event before 8:00AM due to satellite data restriction. As we know that fog may develop further after sunrise. Ignoring the evolution time after 8:00AM may loss many more important features of the fog event, or even the conclusion could be different from this study. Please clarify this concern.

Thanks for this suggestion. We agree that fog would be more complex if it develops after sunrise. We present the number of stations with fog from 07:00AM to 10:00AM (Table X1). At 08:00, the fog begins to dissipate. The number of stations with different VIS levels slightly decreases. Since 08:00, the dissipation process accelerates. At 09:00, there are 40% stations having fog and 20% stations where the VIS is lower than 200m. One hour later, fog remains at only 6 stations. All the fog disappears at 11:00.

Table X1. Number of stations with various VIS levels during 07:00~10:00 on 21 January 2020.

| Local time | Number of stations | VIS<1000m | VIS<500m | VIS<200m |
|---|---|---|---|---|
| 07:00 | | 43 | 28 | 23 |
| 08:00 | 70 | 40 | 27 | 18 |
| 09:00 | | 27 | 17 | 14 |
| 10:00 | | 6 | 4 | 2 |

Therefore, the fog does not develop further after sunrise. It fully dissipates within 3 hours. We believe the fog evolution after 08:00AM has no significant influences on the results at nighttime.

**In Section 3.2**

After 07:00, the fog begins to dissipate, and it fully disappears at 11:00.

4. In the introduction section line 48: Please also add the article by Zhou and Ferrier (2008) about the role of turbulence in fog. Yes, Ye's article (2015) investigated the role of turbulence, but Ye's article is not the first but just confirmed the findings of Zhou and Ferrier in 2008.

Thanks for this suggestion. This article and other related articles have been cited.

Zhou, B. and Ferrier, B.: Asymptotic Analysis of Equilibrium in Radiation Fog, J. Appl. Meteorol, and Climatology, 47, 1704-1722, https://doi.org/10.1175/2007JAMC1685.1, 2008.

Zhou, B. and Du, J.: Fog prediction from a multimodel mesoscale ensemble prediction system, Weather and Forecasting, 25(1), 303-322, https://doi.org/10.1175/2009WAF2222289.1, 2010.

Zhou, B., Du, J., Gultepe, I., and Dimego, G.: Forecast of low visibility and fog from NCEP: Current status and efforts, Pure Appl. Geophys., 169, 895-909, https://doi.org/10.1007/s00024-011-0327-x, 2012.

**In Introduction**

Appropriate turbulence also facilitates fog formation and enhancement (Ye et al., 2015; Zhou and Ferrier, 2008).

**In Section 2.3**

Fog is hard to be simulated or predicted well (Zhou et al., 2010, 2012), which is sensitive to the choice of parameterization schemes (Steeneveld et al., 2014; van der Velde et al., 2010).

---

## Author Comment (AC2)

**Response to Referee#1**

Dear Referee,

Thanks for your constructive comments and suggestions on our manuscript (egusphere-2023-1299). We have studied them carefully and made revisions on the manuscript. These comments and the corresponding replies are listed below.

The reviewer's comments are highlighted by gray. Followed by the comments are our responses. The texts led by **"In Section xx"** are the current texts in manuscript. The added texts or deleted texts are colored by red.

With regards,

Shuqi Yan

**Comments:**

1. Abstract, lines 20/21:" The fog propagation speed would decrease notably by 6.4m/s (66%) if the BLLJ-related moisture and warm advections are turned off." You may add "in the model" to make the concept clearer.

Thanks for this suggestion. We have added "in the model".

> **In Abstract**
>
> The fog propagation speed would decrease notably by 6.4m/s (66%) in the model if the BLLJ-related moisture and warm advections are turned off

2. Suggestion not to use and bold letters/words in the main text body, including abstract.

Thanks for this suggestion. All the bold words are corrected.

3. lines 148 – 151: The directions indicated and angles seem somewhat confusing: 160° means wind from 160° (SE winds) and corresponding fog propagation into the NW direction, correct? Please be more clear.

Thanks for this suggestion. The 160° problem depends on the coordinate system we choose. In the original manuscript, it is in Cartesian coordinate system, not in wind direction coordinate system. It means the direction is from (0,0) to (cos160°, sin160°). The fog propagation is indeed from SE to NW as you mentioned, no matter what coordinate system is chosen. We have added "in Cartesian coordinate system" for clarification.

> **In Section 3.2**
>
> … and the maximum propagation speed is 9.6m/s occurring at 160° direction (in Cartesian coordinate system).

4. Lines 227/228 and other places in the manuscript, including Fig. 13: It seems not quite alright to say that "fog forms at upper level ahead of forming at ground". "Fog" at an upper level with no fog present at the ground, isn't fog! Fog is a cloud with ground contact. If there is no ground contact of the cloud, it is a low stratus cloud. Authors, please revise the manuscript accordingly in order to be more precise with the respective wording. The process that you describe is stratus lowering, as you correctly state in the following..

Thanks for this suggestion. In the first apperance of "upper level fog", we add "The upper-level fog with no ground contact is referred to as low stratus" after it. The related words in the whole manuscript are modified accordingly: "upper level fog"→"low stratus", "subsidence of upper-level fog"→"stratus lowering".

The modifications spread over the manuscript. We only list some representative modifications.

**In Section 3.4**

Additionally seen from Figure 9, the west boundary of vertical fog region below about 100m has a negative slope, i.e., fog forms at upper level ahead of forming at ground. The upper-level fog with no ground contact is referred to as low stratus. The height at which fog/low stratus firstly forms is shown in Figure 10. An initial fog area forms at ground level before 00:00 on 21 January. Since then, low stratus  forms at upper level (about 10~66m) over the downstream area, while the ground fog in downstream area forms about 0~20min later than low stratus . The formation of low stratus  may also be caused by the BLLJ-induced moisture advection. In addition, the  cloud water advection (Section 2.2.3) to downstream area by BLLJ could also be a potential reason. We hypothesize that the formation of ground fog is partly favoured by the stratus lowering , which has been reported by previous studies (e.g., Haeffelin et al., 2010; Liu et al., 2012); the base height of stratus can be smaller than 100m before fog formation (Dupont et al., 2012; Fathalli et al., 2022), which is basically close to our results (10~66m in Figure 10). While in this event, the stratus lowering phenomenon  remains to be verified by additional high-spatiotemporal resolution vertical observations.

According to above results, three potential factors for fog propagation are raised: BLLJ-related temperature advection, moisture advection and  cloud water advection. These advections possibly promote low stratus  formation  within 100m above surface, and subsequently the low stratus  could subside to be ground fog by the turbulent mixing or sedimentation of  cloud droplets. Currently, their contributions to fog propagation have not been quantitatively revealed. Therefore, it will be addressed in the next section.

**In Abstract**

The moisture advection probably promotes  low stratus formation, and later it subsides to be ground fog by turbulent mixing of fog droplets.

Other revisions in Results, Conclusions, figure texts and figure captions are the same with the above.

---

## Author Comment (AC3)

**Response to Referee#2**

Dear Referee,

Thanks for your constructive comments and suggestions on our manuscript (egusphere-2023-1299). We have studied them carefully and made revisions on the manuscript. These comments and the corresponding replies are listed below.

The reviewer's comments are highlighted by gray. Followed by the comments are our responses. The underlined black texts are the original texts in manuscript. The texts led by **"In Section xx"** are the current texts in manuscript. The added texts or  are colored by red. The "Response to Comment n" means that detail information is presented in the response to comment numbered by n.

With regards,

Shuqi Yan

**Comments:**

**1.** First how can be warm moist advection at high levels can generate a fog layer at low levels not clear? It will be lifted with vertical motion to higher levels. A schematic fig shows totally not acceptable conditions. If warm air advection at the surface over the cold surface can generate the fog but not higher levels. What is being told in the paper as stratus formation is possible but not the fog. Warm air adv above generates stable layer, an inversion but not the fog. What will cause the low level fog related jet stream? Did you look at the rad conditions?.

This comment is decomposed into a,b,c.

(a) First how can be warm moist advection at high levels can generate a fog layer at low levels not clear? It will be lifted with vertical motion to higher levels. A schematic fig shows totally not acceptable conditions. If warm air advection at the surface over the cold surface can generate the fog but not higher levels

This question arises from that the heights of warm & moist advection were not stated clearly. In the schematic fig (Fig 13), the arrows of warm & moist advection are plotted above the fog area. In fact, the ocean is warmer and wetter than inland area at nearly all heights below 500m, so warm & moist advection occurs at both surface and higher levels, not merely at the height of the arrows. Of cource the advection is stronger at higher levels. This deep layer of advection is conducive to ground fog formation, which is also observed in many fog cases (e.g., Li et al., 2019; Pu et al., 2008; Wobrock et al., 1992). We add a comment into figure caption: "Note that warm and moisture advections occur at nearly all heights below 500m, not merely at the height indicated by arrows".

**Ref.**

Li, Z., Liu, D., Yan, W., Wang, H., Zhu, C., Zhu, Y., & Zu, F. (2019). Dense fog burst reinforcement over Eastern China: A review. Atmospheric Research, 230(D19), 104639.

Pu, M. J., Zhang, G. Z., Yan, W. L., and Li, Z. H. (2008). Features of a rare advection-radiation fog event. Science China Earth Science, 51(7), 1044–1052.

Wobrock, W., Schell, D., Maser, R., Kessel, M., Jaeschke, W., Fuzzi, S., and Bendix, J. (1992). Meteorological characteristics of the Po Valley fog. Tellus B, 44(5), 469-488.

[Figure]

Figure 13. The concept diagram of fog propagation. The ground wind speed (short orange arrows) is generally less than 3m/s. A southeasterly BLLJ exists at the height from 50 to 500m, and the jet core intensity is 10m/s at 200m (the long orange arrow). The updraft arrows represent the warm and wet air from ocean. The two cloud shapes are fog areas at two adjacent times, and the white arrow indicates the fog propagation speed (9.6m/s). The fog propagation is probably caused by three approaches: 1) Moisture advection from ocean promotes vapor condensation in the downstream area, which could be the dominant cause (the blue fancy arrow); 2) Warm advection from ocean deepens inversion layer and additionally promotes vapor accumulation within PBL (the red fancy arrow); 3) The moisture advection probably result in the low stratus formation, and later it subsides to ground by turbulent mixing of cloud droplets (the blue water drops and circular arrows). Note that warm and moisture advections occur at nearly all heights below 500m, not merely at the height indicated by arrows.

(b) What is being told in the paper as stratus formation is possible but not the fog. Warm air adv above generates stable layer, an inversion but not the fog.

As you mentioned, the warm and moisture advection produces a stable and wet boundary layer. We agree that it definitely has contributions to fog formation and propagation. In the original text, we have stated this opinion in several places, e.g. in *Conclusions*: "1) Moisture advection from ocean promotes vapor condensation in downstream area; 2) Warm advection from ocean deepens inversion layer and additionally promote vapor accumulation within PBL". Nevertheless, the ground wind is small. The advection at ground is conducive to fog propagation, but not enough to explain why the **propagation speed is much faster** than ground wind speed. We argue that the advection could promote fog fast propagation through additional mechanisms. The original text mentioned that "moisture advection could promote fog formation at upper level and then subside to ground". The jet-induced moisture advection favors fog formation at a higher level, and subsequently the upper-level fog subside to be ground fog, which results in fast fog propagation at ground.

The expression of "upper-level fog; upper-level fog subside to ground" are corrected in current version. Strictly speaking, if fog forms at upper level and has not contacted with ground yet, it is called as **low stratus**. The phenomenon of stratus subsiding to be ground fog is referred to as "**stratus lowering**", which is also reported by other studies (e.g., Haeffelin et al., 2010; Dupont et al., 2012). The "upper level fog" in the whole text and figure captions is corrected as "low stratus". Here we only present the revisions in *Section* 3.4. Other revisions are basically the same.

**In Section 3.4**

Additionally seen from Figure 9, the west boundary of vertical fog region below about 100m has a negative slope, i.e., fog forms at upper level ahead of forming at ground. The upper-level fog with no ground contact is referred to

as low stratus. The height at which fog/low stratus firstly forms is shown in Figure 10. An initial fog area forms at ground level before 00:00 on 21 January. Since then, low stratus  forms at upper level (about 10~66m) over the downstream area, while the ground fog in downstream area forms about 0~20min later than low stratus . The formation of low stratus  may also be caused by the BLLJ-induced moisture advection. In addition, the  cloud water advection (Section 2.2.3) to downstream area by BLLJ could also be a potential reason. We hypothesize that the formation of ground fog is partly favoured by the stratus lowering , which has been reported by previous studies (e.g., Haeffelin et al., 2010; Liu et al., 2012); the base height of stratus can be smaller than 100m before fog formation (Dupont et al., 2012; Fathalli et al., 2022), which is basically close to our results (10~66m in Figure 10). While in this event, the stratus lowering phenomenon  remains to be verified by additional high-spatiotemporal resolution vertical observations.

According to above results, three potential factors for fog propagation are raised: BLLJ-related temperature advection, moisture advection and  cloud water advection. These advections possibly promote low stratus  formation  within 100m above surface, and subsequently the low stratus  could subside to be ground fog by the turbulent mixing or sedimentation of  cloud droplets. Currently, their contributions to fog propagation have not been quantitatively revealed. Therefore, it will be addressed in the next section.

**Ref.**

Dupont, J., Haeffelin, M., Protat, A., Bouniol, D., Boyouk, N., and Morille, Y.: Stratus–Fog Formation and Dissipation: A 6-Day Case Study, Boundary-Layer Meteorol, 143, 207–225, https://doi.org/10.1007/s10546-012-9699-4, 2012.

Haeffelin, M., Bergot, T., Elias, T., Tardif, R., Carrer, D., Chazette, P., and Zhang, X.: PARISFOG: Shedding new light on fog physical processes, Bull Am Meteorol Soc, 91(6), 767-783, https://doi.org/10.1175/2009bams2671.1, 2010.

Liu, D. Y., Niu, S. J., Yang, J., Zhao L., Lv, J., and Lu, C.: Summary of a 4-year fog field study in Northern Nanjing, part 1: fog boundary layer, Pure Appl. Geophys, 169, 809–819, https://doi.org/10.1007/s00024-011-0343-x, 2012.

(c) What will cause the low level fog related jet stream? Did you look at the rad conditions?

The jet truly exists during the study period (See Response to Comment3). Other studies also report that boundary-layer low-level jet is a common phenomenon over Yangtze River Delta Region in autumn and winter seasons (Wei et al., 2013). Kraus et al. (1985) have concluded typical reasons for jet formation, such as synoptic systems, topography, inertial oscillation, etc. The possible cause of jet formation has been mentioned in the article: "The formation of BLLJ is likely caused by the easterly movement of a high pressure at 1000hpa over East China. The central pressure gets enhanced, which strengthens the pressure gradient over Jiangsu area and favours wind speed increasing"). Identifying the real cause of jet is not the major concern of this manuscript.

**Ref.**

Wei, W., Wu, B.G., Ye, X.X. et al. Characteristics and Mechanisms of Low-Level Jets in the Yangtze River Delta of China. Boundary-Layer Meteorology, 149, 403–424 (2013).

Kraus, H., Malcher, J. & Schaller, E. A nocturnal low level jet during PUKK. Boundary-Layer Meteorology, 31, 187–195 (1985).

There are three radiation observation stations (Nanjing, Huai'an, Lvsi) in Jiangsu Province. The Huai'an station is covered by this fog event, but longwave radiation data are missing at Huai'an and Lvsi station. We can only look at Nanjing station (Fig X1), which is not far away from fog area. During nighttime, the net radiation is negative due to surface cooling. The simulated radiation flux is close to observation.

[Figure]

Figure X1  The net radiation flux (W/m²) at Nanjing station in nighttime. The black is observation and the red is WRF simulation. The time is from 17:00 Jan20 to 08:00 Jan21.

**2.** Where is the radiative flux in the LWC equation?

The radiative term is implicitly included in the LWC equation.

$$\frac{\partial \text{LWC}}{\partial t} = \underbrace{-\left(u\frac{\partial}{\partial x}+v\frac{\partial}{\partial y}+w\frac{\partial}{\partial z}\right)\text{LWC}}_{\text{Advc}} + \left(\frac{\partial \text{LWC}}{\partial t}\right)_{\text{Vmix}} + \left(\frac{\partial \text{LWC}}{\partial t}\right)_{\text{Cond}} + \left(\frac{\partial \text{LWC}}{\partial t}\right)_{\text{Sedi}} + \left(\frac{\partial \text{LWC}}{\partial t}\right)_{\text{other}}$$

In WRF model, the radiation processes, such as surface longwave radiation, atmospheric longwave radiation and radiative effects of fog droplets, are treated in radiation transfer module. These radiation processes will cause temperature change. The change in temperature is next passed into microphysical module, resulting in condensation/evaporation of fog droplets (the "Cond" term of LWC equation). Therefore, the radiative effects have been already reflected in LWC equation, although in an implicit manner.

**3.** Why the surface wind is less than the higher level as you suggested, and said there are issues in surface obs. I dont think this is true.

We did not mention "there are issues in surface obs" in the whole manuscript. The surface winds are less than 3m/s and higher level winds (50~500m) are larger than 6m/s, which is stated in the original manuscript. The small surface winds and large higher-level winds can be additionally supported by sounding observations. We have added the results of Sheyang station, the nearest sounding station to fog area. Seen from the wind profiles (Fig 6b), the surface winds is small (~2m/s) and higher-level winds (e.g., 200m) is larger than 6m/s, so there are no issues in observations. The observations and simulations at Sheyang station is added into *Section* 3.3.

What we mentioned is "surface wind is small, so it is insufficient to explain fast fog propagation". We indicate that fast fog propagation cannot be explained only by surface winds, not saying that "surface obs having issues".

**In Section 2.1**

This study focuses on the Jiangsu area, China (Figure 1), where a large-scale fog event occurred from 20 to 21 January 2020. We collected the data from 70 ground automatic weather stations (AWS) in Jiangsu Province, China. [data description……]. Additionally, the Sheyang (SY; 120.25 °E, 33.76 °N; 3m) station is a sounding station that used for model evaluation in the vertical direction. The sounding observations include temperature, RH, wind direction and wind speed which are sampled each second. It is conducted twice a day (00UTC and 12UTC).

[Figure]

Figure 1. The parent and nest model domain. The shaded color is terrain height. The red points are automatic weather stations in Jiangsu, China. The three larger circle points are Sihong (SH), Baoying (BY), and Dafeng (DF) stations, and the square point is Sheyang (SY) sounding station. The black labels are some province or city names. (JS:Jiangsu Province; AH:Anhui Province; YC:Yanchen; NT:Nantong).

**In Section 3.3**

Figure 6b evaluates the model performance on temperature, RH and wind field in the vertical direction at SY sounding station. The temperature profile is simulated well by the model, with the mean bias of less than 1K. The RH bias is relatively small below about 200m, while it is a bit larger above 200m at 08:00 on 21 January. The simulated wind speed and direction are basically consistent with observation. The large winds (greater than 6m/s) at about 200m are well reproduced by the model, indicating that the model reasonably simulates boundary layer low-level jet. Studies on boundary layer low-level jet are presented in next sections.

[Figure]

Figure 6b. The model performance on temperature (red), RH (blue) and wind (barbs) profiles at Sheyang sounding station. For temperature and RH, the observations are scatters and simulations are solid lines. For wind barbs, the left column is observations and the right column is simulations. The scatters and barbs are interpolated onto 0~600m every 100m.

**4.** Did you show any obs such as lidar or time series of w at the inversion layer that mixing going on? Say from a turb tower? or aircraft?

There are no w (we think it's vertical wind speed) observations from tower, lidar or aircraft. The simulated vertical wind speed in nighttime is presented in Fig X2. It is commonly small under stable boundary layer conditions (within $\pm 10$ cm/s), since strong vertical motions could not happen in fog events. Other studies also show that vertical wind speed is basically less than 1m/s in fog boundary layer (e.g., Poku et al., 2021; Shen et al., 2022).

**Ref.**

Poku, C., Ross, A. N., Hill, A. A., Blyth, A. M., & Shipway, B. (2021). Is a more physical representation of aerosol activation needed for simulations of fog? Atmospheric Chemistry and Physics, 21(9), 7271–7292.

Shen, P., Liu, D., Gultep, I., Lin, H., Cai, N., & Cao, S. (2022). Boundary layer features of one winter fog in the Yangtze River Delta, China. Pure Appl. Geophys, 179(9), 3463-3480.

[Figure]

Figure X2. Height-time distribution of vertical speed (cm/s) at Dafeng, Baoying and Sihong stations (also the stations used in main text). The time is from 20:00 Jan20 to 08:00 Jan21.

**5.** Did you show a time series of LWC and Nd during the model simulations? I may be missed it.

The simulated LWC and Nd are presented in Figure X3. The times of LWC>0 & Nd>0 and are generally consistent with VIS<1000m, indicating that the model reasonably simulates fog start & end time. The simulated value range are 0~0.6g/kg for LWC and 0~500cm$^{-3}$ for Nd. It is consistent with previous observations (e.g., Gultepe et al., 2006; Haeffelin et al., 2010; Li et al., 2019; Niu et al., 2012).

**Ref.**

Gultepe, I., Müller, M. D., & Boybeyi, Z. (2006). A New Visibility Parameterization for Warm-Fog Applications in Numerical Weather Prediction Models. Journal of Applied Meteorology and Climatology, 45(11), 1469–1480.

Haeffelin, M., Bergot, T., Elias, T., Tardif, R., Carrer, D., & Chazette, P., et al. (2010). Parisfog Shedding new Light on Fog Physical Processes. Bulletin of the American Meteorological Society, 91(6), 767–783.

Li, Z., Liu, D., Yan, W., Wang, H., Zhu, C., Zhu, Y., & Zu, F. (2019). Dense fog burst reinforcement over Eastern China: A review. Atmospheric Research, 230(D19), 104639.

Niu, S., Liu, D., Zhao, L., Lu, C., Lü J., Yang, J. (2012). Summary of a 4-year fog field study in northern Nanjing, part 2: Fog microphysics. Pure and Applied Geophysics, 169(5-6), 1137-1155.

[Figure]

Figure X3  The time series of Vis (observation; black), LWC (simulation; blue) and Nd (simulation; red) at Dafeng, Baoying and Sihong stations (also the stations used in main text). The time is from 14:00 Jan20 to 11:00 Jan21.

**6.** Make comments truly based on models do not prove your hypothesis.

This manuscipt is not truly based on simulations. Generally, all the hypothesis are supported by observation evidence or previous studies.

(a). The WRF model used in this study is reliable. It has been applied to study fog processes by main studies (e.g., Ghude et al., 2023; Steeneveld et al., 2014; Yang et al., 2019; Zhou et al., 2010).

(b). The model reasonably simulates the variation in fog area, surface meteorology and vertical meteorology (Fig 2 and 6 in main text; Response to Comment3). It provides the basis for studying the propagation of fog.

(c). Model simulations show that the jet exists at the height of 50~500m, which may have important effects on fog propagation. The sounding observations also support the existence of jet (Response to Comment3), and the jet speed & direction are consistent with fog propagation indicated by satellite image. The promoting effects of jet on fog are also supported by previous observation or simulation works (Liu et al., 2016; Tian et al., 2019; Wu et al., 2020).

(d). Model simulations indicate that warm and moisture advection contributes to fog formation and propagation. It has been confirmed previously (e.g., Liu et al., 2016; Pu et al., 2008; Wobrock et al., 1992; Wu et al., 2020); the inversion height, inversion intensity and the depth with high vapor content are also consistent with these studies. These supporting papers have been already cited in relevant sections, e.g.:

> **In Section 3.4**
> ......The $Q_v$ under 300m is generally higher than 3g/kg under the effect of BLLJ. Wu et al. (2020) also found that BLLJ continuously transports water vapor to fog layer, resulting in surface $Q_v$ higher than 3g/kg.
>
> ......The approximate inversion layer height is about 100~300m, which is consistent with previous studies (Dorman et al., 2021; Li et al., 2019). The maximum inversion intensity of 15K/100m, which is also reported in a dense fog event (16K/100m) by Pu et al. (2008).

(e). Model simulations indicate that upper-level fog could subside to ground (stratus lowering specifically), which is also reported previously (e.g., Haeffelin et al., 2010; Dupont et al., 2012). In this study, due to the lack of fine observations, we regard it as a potential reason for fog propagation. Therefore, we state "stratus lowering to be ground fog" in a hypothetic tone, not definite tone.

**Ref.**

Dorman, C.E., Hoch, S.W., Gultepe, I., Wang, Q., Yamaguchi, R., Fernando, H., Krishnamurthy, R. (2021). Large-Scale Synoptic Systems and Fog During the C-FOG Field Experiment. Boundary-Layer Meteorology, 181, 171–202.

Dupont, J., Haeffelin, M., Protat, A., Bouniol, D., Boyouk, N., and Morille, Y. (2012). Stratus–Fog Formation and Dissipation: A 6-Day Case Study. Boundary-Layer Meteorology, 143, 207–225.

Ghude, S. D., Jenamani, R. K., Kulkarni, R., Wagh, S., Dhangar, N. G., & Parde, A. N., et al. (2023). WiFEX: Walk into the Warm Fog over Indo-Gangetic Plain Region. Bulletin of the American Meteorological Society, 104(5), E980-E1005.

Haeffelin, M., Bergot, T., Elias, T., Tardif, R., Carrer, D., Chazette, P., and Zhang, X. (2010). PARISFOG: Shedding new light on fog physical processes. Bulletin of the American Meteorological Society, 91(6), 767-783.

Liu, D., Yan, W., Yang, J., Pu, M., Niu, S., Li, Z. (2016). A Study of the Physical Processes of an Advection Fog Boundary Layer. Boundary-Layer Meteorology, 158(1), 125-138.

Pu, M. J., Zhang, G. Z., Yan, W. L., and Li, Z. H. (2008). Features of a rare advection-radiation fog event. Science China Earth Science, 51(7), 1044–1052.

Steeneveld, G. J., Ronda, R. J., Holtslag, A. A. M. (2014). The Challenge of Forecasting the Onset and Development of Radiation Fog Using Mesoscale Atmospheric Models. Boundary-Layer Meteorology, 154(2), 265–289.

Tian, M., Wu, B., Huang, H., Zhang, H., Zhang, W., and Wang, Z. (2019). Impact of water vapor transfer on a Circum-Bohai-Sea heavy fog Observation and numerical simulation. Atmos. Res., 229, 1-22.

Wobrock, W., Schell, D., Maser, R., Kessel, M., Jaeschke, W., Fuzzi, S., and Bendix, J. (1992). Meteorological characteristics of the Po Valley fog. Tellus B, 44(5), 469-488.

Wu, B., Li, Z., Ju, T., and Zhang, H. (2020). Characteristics of Low-level jets during 2015–2016 and the effect on fog in Tianjin. Atmos. Res., 245, 105102.

Yang, Y., Hu, X.-M., Gao, S., & Wang, Y. (2019). Sensitivity of WRF simulations with the YSU PBL scheme to the lowest model level height for a sea fog event over the Yellow Sea. Atmospheric Research, 215, 253–267.

Zhou, B., & Du, J. (2010). Fog Prediction from a Multimodel Mesoscale Ensemble Prediction System. Weather and Forecasting, 25(1), 303–322.

**7.** Some conclusions are very vague, needs to be supported. I see also no discussion section.

Thanks for this suggestion. The vague conclusions mentioned in Comment 1~6 to are made more clear. Here we summarize our responses again as follows:

(a) The relations between warm & moisture advection and fog

The warm & moisture advection occurs at both surface and higher levels, not merely at the height of the arrows in schematic fig. So it is definitely conducive to ground fog formation. It is now clarified in the manuscript (See Response to Comment 1).

(b) Concerns about stratus/upper-level fog

The ground wind is small (<3m/s), so the warm & moisture advection at ground may be not enough to explain why the propagation speed is so fast (~10m/s). We argue that moisture advection could promote fog formation at upper level (low stratus) and then subside to ground (stratus lowering). The "upper-level fog" in the whole text is corrected as "low stratus" (See Response to Comment 1).

(c) Issues about surface wind observation and jet

Ground wind is small and upper-level (50~500m) wind is large. This is a true phenomenon, having no issues. The existence of jet is supported by sounding observations (See Response to Comment 3).

(d) "Make comments truly based on models do not prove hypothesis".

Nearly all the hypothesis have been supported by observation evidence or previous studies (Response to Comment 6).

(e) "I see also no discussion section".

Although *Discussion* is not compulsory in ACP, we now add this section. Since the last two paragraphs in *Results* summarize the story of fog propagation and provide the implications in a broader view, we reorganize them into *Discussion*.

**Discussion**

Previous studies have elucidated the qualitative reasons for fog propagation. In this study, we describe the feature of fast fog propagation and identify its key impact factors more quantitatively. Figure 13 summarizes the mechanism of fog propagation. During the nighttime, a southerly BLLJ controls the study region, and the jet core intensity is about 10m/s which occurs at about 200m. The ground fog propagates northwestward with the speed of 9.6m/s. The BLLJ favours the fast fog propagation by three possible mechanisms: 1) BLLJ transports sufficient vapor from ocean to inland area. The turbulence strengthened by wind speed shear further moistens the PBL and promotes vapor condensation. This could be the dominant mechanism. 2) BLLJ transports warmer air from ocean to inland area and deepens the inversion layer. The strengthened inversion favours the accumulation of vapor and condensation nuclei. 3) The strong moisture advection could promote the low stratus formation in the downstream area, and later it subsides to be ground fog by turbulent exchange of  cloud droplets. The stratus lowering phenomenon  needs to be verified by additional observations.

The results could facilitate the understanding of cloud formation and development. Clouds, such as convective clouds, can develop and expand extraordinarily fast under strong synoptic forcing or unstable conditions. Fog can be viewed as a kind of near-surface stratus cloud, which usually forms under stable conditions with weak synoptic forcings. However, as revealed in this study, it can also develop and propagate fast under the effect of BLLJ. The quantitative relations between BLLJ and fog fast propagation may have implications on the cloud formation and development mechanism under stable synoptic conditions.